# Therapeutic Potential of an Azithromycin-Colistin Combination against XDR *K. pneumoniae* in a 3D Collagen-Based In Vitro Wound Model of a Biofilm Infection

**DOI:** 10.3390/antibiotics12020293

**Published:** 2023-02-01

**Authors:** Olena V. Moshynets, Taras P. Baranovskyi, Olga S. Iungin, Alexey A. Krikunov, Viktoria V. Potochilova, Kateryna L. Rudnieva, Geert Potters, Ianina Pokholenko

**Affiliations:** 1Biofilm Study Group, Department of Cell Regulatory Mechanisms, Institute of Molecular Biology and Genetics, National Academy of Sciences of Ukraine, Zabolotnoho Str. 150, 03680 Kyiv, Ukraine; 2CeMM, Research Center for Molecular Medicine of the Austrian Academy of Sciences, Lazarettgasse 14, A-1090 Vienna, Austria; 3Department of Biotechnology, Leather and Fur, Faculty of Chemical and Biopharmaceutical Technologies, Kyiv National University of Technologies and Design, Nemyrovycha-Danchenka Street 2, 01011 Kyiv, Ukraine; 4National Amosov Institute of Cardio-Vascular Surgery Affiliated to National Academy of Medical Sciences of Ukraine, Amosov Str. 6, 02000 Kyiv, Ukraine; 5Kyiv City Maternity Hospital № 2, Mostytska 11, 02000 Kyiv, Ukraine; 6Kyiv Regional Clinical Hospital, Baggovutovskaya Str. 1, 04107 Kyiv, Ukraine; 7Department of Microbiology, Virology and Immunology, Bogomolets National Medical University, Shevchenka Blvd. 13, 01601 Kyiv, Ukraine; 8Antwerp Maritime Academy, Noordkasteel Oost 6, 2030 Antwerp, Belgium; 9Department of Bioscience Engineering, University of Antwerp, Groenenborgerlaan 171, 2020 Antwerp, Belgium; 10Department of Cell Regulatory Mechanisms, Institute of Molecular Biology and Genetics, National Academy of Sciences of Ukraine, 150 Zabolotnoho Str., 03680 Kyiv, Ukraine; 11The Polymer Chemistry & Biomaterials Group, Department of Organic and Macromolecular Chemistry, Ghent University, Krijgslaan 281, S4-Bis, 9000 Ghent, Belgium

**Keywords:** MDR/XDR/PDR Gram-negative infection, *Klebsiella pneumonia*, biofilms, wound model, colistin methanesulfonate, azithromycin, combined antibacterial therapy

## Abstract

A therapeutic combination of azithromycin (AZM) and colistin methanesulfonate (CMS) was shown to be effective against both non-PDR and PDR *Klebsiella pneumoniae* biofilms in vitro. These anti-biofilm effects, however, may not correlate with effects observed in standard plate assays, nor will they representative of in vivo therapeutic action. After all, biofilm-associated infection processes are also impacted by the presence of wound bed components, such as host cells or wound fluids, which can all affect the antibiotic effectiveness. Therefore, an in vitro wound model of biofilm infection which partially mimics the complex microenvironment of infected wounds was developed to investigate the therapeutic potential of an AZM-CMS combination against XDR *K. pneumoniae* isolates. The model consists of a 3D collagen sponge-like scaffold seeded with HEK293 cells submerged in a fluid milieu mimicking the wound bed exudate. Media that were tested were all based on different strengths of Dulbecco’s modified Eagles/high glucose medium supplemented with fetal bovine serum, and/or Bacto Proteose peptone. Use of this model confirmed AZM to be a highly effective antibiofilm component, when applied alone or in combination with CMS, whereas CMS alone had little antibacterial effectiveness or even stimulated biofilm development. The wound model proposed here proves therefore, to be an effective aid in the study of drug combinations under realistic conditions.

## 1. Introduction

Lately, difficult-to-treat multi-drug resistant (MDR), extensively drug-resistant (XDR) and pandrug-resistant (PDR) gram-negative bacterial (GNB) infections are occurring more frequently all over the world, putting pressure on health care personnel to use so-called last-resort antibiotic compounds. Different forms of polymyxins belong to this group of last-resort agents. These compounds had been in use until the 1980s, but over the last decades, they were recently put forward again, in response to increased numbers of MDR/XDR/PDR GNB-associated infections [1,2,3]. 

In particular, two pharmacological types of these therapeutic polymyxins are currently in use as last-resort antibacterial agents for the treatment of XDR and PDR GNB strains: one of them is colistin sulfate (COL), and the other is colistin methanesulfonate (CMS, also known as colistimethate sodium), an improved version of colistin [4,5]. CMS is considered a prodrug of COL and transforms, upon uptake into an organism, into an active polymyxin cation, though with an unknown ratio and uncertain pharmacodynamics and with low effectiveness [4,5,6]. Generally, CMS might be considered a “weak antibiotic” with a 28-day patient mortality of about 43% [7]. On the other hand, CMS is less toxic to patients compared with COL; however, the compound is also less effective against bacteria and has poor bioavailability, i.e., low concentration in the pleural cavity, bronchoalveolar lavage (BAL), lung parenchyma, bones, and the cerebrospinal fluid, where the colistin distribution may correspond to 15–25% of the plasma concentration or even remain undetectable [5,6,8].

Nevertheless, the effectiveness of even these weaker antibiotics may be synergistically enhanced when applied therapeutically in combination with antibiotics with a different mode of action. Macrolides, for example, may present themselves as an option for a combined treatment with CMS or COL. While they are usually not a proper option for the treatment of GNB, because these bacteria are considered inherently resistant against this class of antibiotics [9], a combination of CMS and azithromycin (AZM) has been shown to have a synergistic antibacterial effect on both planktonic growth and biofilm development of a standard *K. pneumoniae* ATCC 10031 culture, in a physiological concentration range [10]. Moreover, this holds for non-MDR, MDR, and colistin-resistant PDR *K. pneumoniae* hospital isolates as well [10]. 

Interestingly, this synergy between CMS and AZM in a standard agar-diffusion assay did not match the effects observed in the corresponding biofilm models, possibly because biofilms are natural aggregates while agar colonies are artificial ones due to the unnatural conditions under which they are grown [11]. One might even say that the existence of such discrepancies suggests that disc-diffusion and broth growth-based assays may not be good predictors of antibiotic susceptibility in biofilms, and that biofilm-based tests are better models to evaluate antibiotic effectiveness. Moreover, even biofilms are not the best in vitro test systems to test antibiotic effectiveness, since they lack the eukaryotic (host) component of the infection. Both the microbial pathogens and key wound bed components (an extracellular matrix, the presence of cells from the host and the fluids in the wound milieu) are critical, each in their own specific way, for biofilm-associated infection development. These factors should therefore be taken into account when biomimetic in vitro models are being studied [12].

One of these components is the extracellular matrix protein collagen. It is the most abundant component of the extracellular matrix and provides tensile strength, regulates cell adhesion, supports chemotaxis and migration, and directs tissue development [13]. At present, collagen can be extracted from various natural animal sources or be obtained from recombinant production systems in bacteria, plants, etc. [14,15,16,17]. Due to the role of the extracellular matrix in infection development, collagen-based gel matrices have been used extensively as a substrate for the in vitro culture of biofilms [18]. Secondly, the introduction of host cells into the model allows for a better description of the mechanisms of the interactions between the cells and the bacteria, as well as of the involvement in biofilm development and protection of the bacteria from being killed by antibiotics. Thirdly, the medium environment used in the model to reproduce wound fluid milieu, affects antibiotic behavior. For instance, AZM seemed to have a stronger antibacterial effect on some GNB in eukaryotic media [9,19,20,21]. This might be associated with the cell wall rebuilding due to OprM downregulation [19] which may potentially improve the antibacterial effects of other antibiotics used in combination with AZM [20,21]. This might also influence the CMS susceptibility in a biofilm culture, since a synergistic effect between CMS and AZM was already observed [10,20]. 

To produce a better model to evaluate the treatment efficiency of a combination of antibiotics against wound infections, three biomimetic wound surface models were constructed with collagen-based 3D scaffolds and seeded with epithelial-like cells. These wound surface models were then used to investigate the therapeutic potential of CMS-AZM combinations against ХDR *K. pneumoniae* isolates. 

## 2. Results

### 2.1. The Influence of Different Culture Media and the Presence of HEK293 on the Growth of K. pneumoniae UHI 1090

In order to construct a working model imitating a wound, both the scaffolding on which the cells can develop, and the liquid phase, which mimics the wound fluids, should be chosen carefully. The liquid phase should contain serum, red blood cells, plasma, and a source of hydrolyzed proteins, such as brain heart infusion media or peptones [22]. At the same time, the medium components should potentiate microbial growth as well as biofilm formation and should not demonstrate any inhibitory effect. To evaluate this, different modifications of the liquid phase content have been studied, following their effects on the *K. pneumoniae* UHI 1090 culture. A total of three modifications of the general growth medium of the bacterial culture were chosen to be seeded with or without the mammalian HEK293 cells: (1) 90% DMEM/high glucose, 10% fetal bovine serum (FBS); (2) 89.9% DMEM/high glucose, 10% FBS, 0.01% Bacto Proteose Peptone; (3) 49.9% DMEM/high glucose, 50% FBS, 0.1% Bacto Proteose Peptone. The data obtained revealed that all culture media supported the growth of *K. pneumoniae* UHI 1090 (Figure 1). An increase in optical density during 24 h in culture was more pronounced in variants that contained the HEK293 cells (Figure 1a).

Addition of HEK293 cells to each of the media promoted microbial growth within 24 h of cultivation (Figure 1a) and even within 48 h for the third medium (49.9% DMEM/high glucose, 50% FBS, 0.1% Bacto Proteose Peptone) (Figure 1b). The high FBS content in the third medium emulates the bed wound exudate, the presence of which can stimulate bacterial culture development. As demonstrated previously, *K. pneumoniae* may induce cytotoxic effects in lung epithelial cells in vitro, as a kind of predatory effect, and the cytotoxicity was highly dependent on the presence and properties of capsule polysaccharides [23]. Since *K. pneumoniae* UHI 1090 is a hospital pathogenic isolate, it is quite possible that it has similar cytotoxic effects on the HEK293 cells. Also, this third culture medium has the highest content of proteins as well as their hydrolysis products, and the presence of HEK293 cells was linked to the most pronounced stimulation of bacterial growth during 48 h in culture. Since all media tested here showed good microbial tolerance, these culture media were selected for further studies. 

### 2.2. AZM and CMS Effects on the Viability of HEK293 Cells In Vitro

Some antibiotics may not only influence bacterial cells but have also a general toxic effect on eukaryotic cells by causing mitochondrial dysfunction, or by generating abnormally high levels of oxygen reactive species, which leads to oxidative damage to the cells [24]. This prompted the need to study the effects of AZM and CMS on the metabolic activity of the HEK293 cells (Figure 2).

CMS seems to decrease the metabolic activity of the HEK293 cells by 17–34% in a range from 1 to 50 mg/L compared to the non-treated controls (Figure 2a). As the HEK293 cell line was derived from the human embryonal kidney, this cytotoxicity of the CMS can be associated with the nephrotoxicity known to be induced by polymyxins [25,26]. AZM also exhibited cytotoxic effects towards HEK293, however, in a clear dose-dependent manner. In concentrations of 20–90 mg/L, it induced a 9–44% reduction in the cell’s metabolic activity with the maximum effect at the highest dose tested (Figure 2b), while at 10 mg/L of AZM, the lowest concentration in the test, it did not cause any significant decrease in metabolic activity.

According to pharmacokinetic literature, the maximum serum concentration of CMS (C_max_) corresponds to 10–12 mg/L [27,28]. Therefore, only concentrations lower than this C_max_ should be tested, as higher concentrations are not physiologically relevant. Serum C_max_ for AZM varies from 1 to 9 mg/L with tissue concentrations 10–50-fold higher and a 24–34% increase in infected/inflamed tissues [29,30]. Our cytotoxicity assays, however, suggest that 5 mg/L of CMS and 10 mg/L of AZM, which both fall into the physiological range, would not impose a significant effect on the eukaryotic cell viability (Figure 2). These amounts could therefore, be expected in patients’ sera and tissues following a standard dosage regime.

### 2.3. Development and Characterization of a Porous Collagen Scaffold for a 3D Collagen-Based Wound Model In Vitro

A porous 3D collagen-based matrix colonized with the epithelial-like HEK293 cells, and immersed in culture media that mimic wound exudate, has been selected to develop an in vitro model which maximally mimics the tissue-specific parameters during a wound-like infection during which bacterial biofilms form on the soft tissues. Type I collagen was selected for the development of the 3D scaffold because it is one of the main components of the derma. The matrices were developed by freeze-drying a solution of bovine collagen type I. The data received by SEM revealed that scaffolds had pores with an average pore diameter of 117 ± 57 µm (Figure 3). The internal microstructure of the scaffolds consisted of multi-layered porous sheets (as seen from the cross-section SEM image in Figure 3a) with an average distance between the layers of 91 ± 50 µm. The porous structure of the developed 3D scaffolds facilitates cell migration inside the scaffold, ensures the exchange of the culture medium as well as gases and allows for the removal of cell metabolism products.

This collagen scaffold was tested to see whether it could be colonized by the epithelial-like cell line HEK293, as well as by a XDR hospital isolate of *K. pneumoniae* UHI 1090. To evaluate whether HEK293 could colonize the developed collagen scaffold, 2 × 10^5^ HEK293 cells were seeded onto a collagen scaffold of approximately 0.024 сm^3^ in volume, immersed in DMEM/high glucose medium containing 10% FBS. After three days, colonization could be confirmed (Figure 4).

Interestingly, *K. pneumoniae* UHI 1090 failed to develop biofilms on this collagen scaffold if it was not colonized with HEK293 (Figure 5). Briefly, when 10^4^ CFU/mL of the bacterial culture was inoculated, there were no bacterial biofilms observed following 48 h of incubation. Instead, individually attached bacterial cells were found on the collagen surface, and no CFU was found in the liquid medium, suggesting that no planktonic bacterial subpopulation had developed. 

Thus, to mimic the conditions under which these can be observed in vivo, the presence of eukaryotic cells might be specifically important to allow pathogenic isolates to develop both their biofilms and planktonic subpopulations. 

### 2.4. AZM and CMS Effects on Biofilm Development by K. pneumoniae UHI 1090 in a 3D Collagen-Based In Vitro Model

The first model, developed in medium No. 1, demonstrated well-developed biofilms in a control where no eukaryotic cells remained and even better biofilm biomass where CMS was added (Figure 6). The level of CFU in the liquid medium phase correlated with the biofilm biomass present and was 10^7^ for control and 10^8^ CFU/mL for the CMS microcosm correspondingly, which demonstrated a stimulatory effect of CMS instead of the expected antibacterial action. In the AZM-containing microcosm, the *Klebsiella* CFU was 10^2^ per mL and the eukaryotic cells were mostly preserved and the biofilms did not develop with only one field of view where a few attached bacterial cells were found (Figure 6, right image). Both antibiotics had strong antibacterial effects and inhibited the development of the biofilm on the wound model, however, with ten bacterial cells/mL as a planktonic subpopulation. 

Interestingly, bacterial functional amyloids were observed to be associated with *Klebsiella* cells when medium No. 1 was applied (Figure 7). Previously, we demonstrated that AmyGreen visualizes functional bacterial amyloids more effectively than the classical Thioflavine T [31,32]. However, amyloidogenesis was repressed when AZM was used. 

The second model was based on medium No. 2. This model proved to be less effective than the previous one although neither medium No. 1 nor No. 2 demonstrate any differences in the plate-based assay (Figure 8). Upon application of medium No. 2, biofilms were only observed under control conditions, where no antibiotics had been added. The corresponding planktonic subpopulations were also suppressed: the control had 10^5^ CFU/mL, whereas no CFU were observed in antibiotic-containing variants. 

Medium No. 3 was used in the third model. Like in the first model, biofilm development and absence of eukaryotic nuclei were observed in control and CMS-containing variants when AZM and AZM + CMS variants demonstrated partially preserved initial model structure with some individual bacterial cells attached (Figure 9). Plating assays confirmed the cytological observations: planktonic cell concentrations of 10^8^ CFU/mL were observed in control and CMS-supplemented variants, while concentrations of 10^2^ CFU/mL were observed in AZM and AZM + CMS variants, which suggests that CMS did not show any antibacterial effect even though *K. pneumoniae* UHI 1090 was sensitive to polymyxin. On the contrary, AZM demonstrated strong antibacterial effects against planktonic and biofilm subpopulations.

## 3. Discussion

Polymyxin antibiotics demonstrate an extremely narrow therapeutic index of 2–4 mg/L [33]. This reduces the application of colistin considerably: an antibacterial effect may overlap with the nephrotoxic dose regimen [26]. Thus, since the introduction of polymyxins in the 1950s, the most effective therapeutic regime has been debated [8,33,34]. As the overall success of CMS therapy stands at approximately 50%, there is a prediction that better balanced regime might improve the therapeutic outcome. This can be achieved by increasing the C_max_ up to the MIC and beyond. However, increasing the dosage does not help to achieve this goal. Even 3 million units (MU) i/v every 8 h does not provide a needed C_max_:MIC ratio for COL-susceptible GNB with MIC ≤ 2 (2022 EUCAST Clinical Breakpoints; European Committee on Antimicrobial Susceptibility Testing). Particularly, 2 MU of CMS i/v every 8 h resulted in a COL concentration of 0.92 ± 0.46 mg/L or 1.03 ± 0.69 mg/L in another study, when 3 MU i/v every 8 h corresponded to 0.6–2.3 mg/L of COL [8,27,35]. In another study, the C_max_ of COL concentration in plasma following i/v administration of 3 MU of CMS every 8 h, 4.5 MU every 12 h and 9 MU every 24 h corresponded to 3.34 +/− 0.35, 2.98 +/− 0.27 and 5.63 +/−0.87 mg/L, respectively [36]. Moreover, all serum samples containing more than 4 mg/L COL eliminated *P. aeruginosa,* whereas complete bacterial eradication was only achieved in 40% of the samples containing less than 4 mg/L COL [36]. Since the MIC breakpoint for COL-susceptible GNB corresponds to 2 mg/L, the widely used dosage regimes of CMS are associated with suboptimal C_max_:MIC ratios, which still need to be determined. Still, based on [36] we can conclude that the empirical C_max_:MIC-ratio would have to be more than two, or that C_max_ has to be more than 4 mg/L for GNB. Taking this into account, several approaches for a colistin dosing regimen have been adopted, such as the application of a loading dose followed by a maintenance dose, a higher dose as per patient renal function and targeted steady-state concentrations of colistin, local administration (intraventricular or inhaled), and antibiotic combination therapy [37]. Following their systematic review, Haseeb et al. [33] concluded that a loading dose of 9 MU of CMS followed by a maintenance dose of 4.5 MU every 12 h was considered the most appropriate dosing strategy to optimize the safety and efficacy of treatment and improve clinical outcomes. Katip et al. [38] also concluded that the loading dose of 9 MU should be used to increase the opportunity for patients to achieve favorable outcomes. However, an increase in nephrotoxicity was found to be associated with the 9 MU loading dose application [38].

Another way to optimize the CMS therapy might be an approach to decrease the MIC of COL/CMS by applying a synergistic antibiotic with another mechanism of action, presumably one which can decrease metabolic activity and biofilm formation to reduce phenotypic resistance. Macrolides correspond to such requirements. Even though none of the macrolides are officially recommended to treat Gram-negative infections due to their natural genetic resistance, there is some evidence that some macrolides have anti-biofilm and anti-virulence effects on *P. aeruginosa* [9,39,40,41,42], which could be associated with protein synthesis inhibition due to the formation of a ribosomal-macrolide complex [43,44]. This results in an inhibition of the general metabolic activity of a cell albeit without a lethal effect due to drug efflux. Thus, any of the macrolides with the same mode of action may be effective in metabolic suppression and, as a result, have anti-biofilm efficacy. Indeed, two out of eight macrolides repressed biofilm formation by *K. pneumoniae,* with azithromycin effectively suppressing ATCC, hospital acquired MDR/PDR and non-MDR *K. pneumoniae* isolates [10]. It was quite surprising not to see both erythromycin and clarithromycin as effective anti-biofilm macrolides, as both antibiotics were described as effective as azithromycin in the treatment of *P. aeruginosa*-related infections [44,45,46,47,48]. Also, it was unexpected to find josamycin as effective as azithromycin to suppress *K. pneumoniae* biofilm development as several studies reported that josamycin was absolutely not effective against *P. aeruginosa* [44,48,49]. That might suggest that effects of different macrolides on Gram-negative bacteria are independent of the number of lactone rings, as was suggested by the above-mentioned authors, but are, on the contrary species dependent. 

Recently we have investigated and described a synergistic effect of CMS and AZM applied together against planktonic populations and biofilms of XDR/PDR and non-MDR *K. pneumoniae* hospital isolates [10]. In order to investigate how the same combination of antibacterial compounds can act against XDR (sensitive to colistin) *K. pneumoniae* infection, three modifications of the in vitro wound model based on a collagen scaffold seeded with epithelial-like cell line HEK293 were used. All the models differed only with respect to the liquid phase in which the scaffold with eukaryotic cells was submerged. Even though none of the liquid phases showed significant differences in their ability to support bacterial growth, the models were considerably different in terms of bacterial behavior and antibiotic effectiveness. The second model was less favorable for the modulation of the infection process since the control variant (no antibiotic added) demonstrated weak bacterial growth (10^5^ CFU/mL). The first and the third models were more effective: the first model was able to stimulate the appearance of bacterial amyloid fibrils, and both of the models confirmed the absence of antibacterial activity of CMS (in a 5 mg/L concentration), as was expected. 

Intriguingly, as was already discussed by [9], a significant difference was observed in the effects of AZM and CMS on *K. pneumoniae* UHI 1090, where 9 mg/L of AZM was able to reduce biofilm development by 7% in a *K. pneumoniae* UHI 1090 biofilm model (Figure 10, adapted from Figure 5 in [10]) and the in vitro wound model used in this study.

In our in vitro wound model, AZM considerably suppressed the biofilm formation and reduced CFU from 10^7^–10^8^ CFU/mL up to 10^2^ CFU/mL, which confirmed a considerable improvement of antibacterial efficiency of AZM against GNB in the physiologically relevant environment as observed before [9,19,20,21]. Moreover, the synergistic action of AZM and CMS observed in a biofilm model, as mentioned before (Figure 11, [10]), also considerably improved. Briefly, a combined application of 8 mg/L CMS and 9 mg/L AZM reduced biofilm development by 18% compared to the control. In our models, lower concentrations of CMS (5 mg/L) and 10 mg/L AZM, demonstrated a bactericidal effect in model No. 1 and a considerable bacterial suppression in model No. 3, illustrated by weak bacterial growth (100 CFU/mL) and a partial retention of the model structure. 

Analysis of the effects of AZM in model No. 1 and model No. 3 demonstrates a strong bactericidal effect of AZM against XDR *K. pneumoniae* UHI, which was not observed in a biofilm model used before [10]. On the contrary, CMS did not demonstrate any antibacterial (model No. 3) or even a growth-stimulating effect (model No. 1). It is well known that CMS interacts with the negatively charged phosphate groups of Lipid A on the outer membrane of gram-negative bacteria [50], leading to the impairment of the three-dimensional LPS structure. However, the interaction of LPS with the chromosomal DNA that was released from the dying HEK293 cells could possibly block the binding of colistin to its cellular target. Also, DNA can serve as a scavenger for the CMS molecules. It was demonstrated previously that polymyxin B could bind the chromosomal DNA of mammalian cells [51]. The increased bacterial growth on the less nutritious culture medium 1 in the presence of CMS could be explained by the early release of additional nutrients from HEK293 stimulated by the cytotoxic action of CMS on these cells. On the other side, sub-MIC concentrations of CMS were shown to enhance biofilm formation by upregulating 197 genes responsible for the formation of extracellular polymeric substances, the production of homoserine lactone and the expression of amyloid, as well as to slow down cell metabolism by downregulating 88 genes in *Acinetobacter baumannii* [52]. Both the stimulation effect as well as the enhancement of amyloid production observed in Figure 7 might therefore be a direct response to a sub-MIC regime of CMS. 

## 4. Materials and Methods

### 4.1. Bacterial Strain and Culture Conditions

*Klebsiella pneumoniae* UHI 1090, a Ukrainian hospital isolate, was originally recovered from a patient with enteritis and identified as an extensively drug-resistant (XDR) strain using antibiotic disc diffusion assays and EUCAST 2021 v.11.0 breakpoints [10]. An AST-N332 card was used to confirm the XDR strain phenotype with the VITEK 2 Advanced Expert System. Antimicrobial susceptibility to polymyxin was tested with a SensiTest Colistin (Liofilchem, Roseto degli Abruzzi, Italy) broth microdilution assay and the results were interpreted according to EUCAST breakpoints. The strain was found to be sensitive to polymyxin B and tigecycline but resistant to all β-lactams, fluoroquinolones and aminoglycosides (details of which can be found at https://doi.org/10.1371/journal.pone.0270983.s006) (accessed on 28 December 2022). The strain was stored as frozen stocks in 25 (*v*/*v*) % glycerol at −80 °C.

### 4.2. Mammalian Cell Lines

The human embryonal kidney cell line HEK293 was obtained from the Bank of Cell Lines from Human and Animal Tissues of the Kavetsky Institute of Experimental Pathology, Oncology and Radiobiology of the National Academy of Science of Ukraine. The cells were cultured in Dulbecco’s modified Eagles medium (DMEM)/high glucose culture medium (Biowest, Nuaillé, France), containing 10% fetal bovine serum (FBS) (Sigma-Aldrich, Saint Louis, MO, USA), 100 U/mL of penicillin, and 100 mg/mL of streptomycin (Arterium, Kyiv, Ukraine).

### 4.3. Viability of Eukaryotic Cells after Antibiotic Treatment

The influence of CMS and AZM on the viability of HEK293 cells was tested with the colorimetric MTT metabolic activity assay. HEK293 cells were seeded 24 h prior to the application of the antibiotics. Before any antibiotics were added, the culture medium was changed to a medium without fetal bovine serum. HEK293 cells were then cultured at 37 °C and 5% CO_2_, at a concentration of 3 × 10^5^ cells per well in a 96-well plate, in the presence of varying concentrations of the antibiotic for 48 h. Cells grown in DMEM/high glucose medium without antibiotics served as a negative control group. The assay was performed according to [53]. Measurements for each concentration point were performed in triplicate. 

### 4.4. Effects of Different Culture Media and HEK293 Cells on the Growth of K. pneumoniae UHI 1090

A total of 3 × 10^5^ HEK293 cells/well were plated into 96-well culture plate 24 h before bacterial inoculation. The HEK293 cells were cultivated in DMEM/high glucose medium, containing 10% FBS at 37 °C, 5% CO_2_. The medium was changed before inoculation of *K. pneumoniae* to the following media: (1) 90% DMEM/high glucose medium, containing 10% FBS; (2) 89.99% DMEM/high glucose medium, containing 10% FBS, 0.01% Bacto Proteose Peptone (BD Biosciences, Franklin Lakes, NJ, USA); (3) 49.99% DMEM/high glucose medium, containing 50% FBS, 0.01% Bacto Proteose Peptone (BD Biosciences, Franklin Lakes, NJ, USA). Cell-free media served as control. *K. pneumoniae* UHI 1090 (10^4^ CFU/well) was inoculated, and the incubation was prolonged at 37 °C, 5% CO_2_. OD values were measured 24 h after inoculation at 620 nm. Each measurement was performed in six replicates. 

### 4.5. Cultivation of Bacteria and Antibiotic Treatments

The bacterial strains were cultured aerobically at 37 °C in Luria–Bertani (LB) medium (10 g/L peptone, 5 g/L yeast extract and 10 g/L sodium chloride, with 12 g/L agar added when solid media were needed [54]) and on Muller Hinton Agar (OXOID, Basingstoke, UK). The strain was recovered from frozen stocks on LB plates before initiating overnight shaken cultures to provide fresh inoculum for experiments (direct inoculation from frozen stocks was found to reduce siderophore production and biofilm development considerably; unpublished observations). Culture densities and dilutions were determined by OD 570 measurements using a Multiskan™ FC Microplate Photometer (Thermo Fisher Scientific, Waltham, MA, USA). A 10 g/L AZM (Pharmex Group, Boryspil, Ukraine) stock solution was prepared using DMSO as a solvent, while a 1 g/L stock solution of CMS (Forest Laboratories, Barnstaple, UK) was prepared in water. Both stocks were used within 30 min of preparation. Discs containing antibiotics (HiMedia Laboratories, Mumbai, India) as well as AST-N332 cards for the VITEK 2 Advanced Expert System (bioMerieux, Marcy l’Étoile, France) were used for antimicrobial susceptibility and sensitivity assays as listed in [10].

### 4.6. Collagen Scaffold Preparation

Porous collagen scaffolds were produced from a solution of bovine atelocollagen in acetic acid using the following freeze-drying technique. Briefly, a solution of 20 mg/mL of type I atelocollagen, isolated from a bovine tendon, was prepared in 0.5 M acetic acid. Afterwards, the solution was centrifuged at 2500 rpm, +4 °C for 15 min to remove air bubbles formed during mixing. The prepared collagen solution was then frozen in glass Petri dishes (10 cm in diameter) at −40 °C and held for 18 h inside a freeze-dryer. The frozen suspensions were subsequently sublimed at −40 °C to +22 °C for 24 h under a vacuum. The atelocollagen scaffolds were stored at −20 °C until further use. Prior to usage, the scaffolds were brought to +22 °C, cut into 3 × 4 mm rectangles under a sterile laminar hood, and sterilized by UV-exposure for 40 min. Subsequently, they were equilibrated in 0.1 M HEPES (pH 8.0) at +4 °C overnight. Afterwards, the HEPES solution was changed with DMEM/high glucose medium, containing 10% (*v*/*v*) FBS, 100 U/mL penicillin and 100 µg/mL streptomycin, in which the scaffolds were equilibrated for 48 h, at +37 °C and 5% CO_2_, before cell seeding.

### 4.7. Preparation of the Wound Model 

HEK293 cells were grown in Dulbecco’s modified Eagle’s medium (DMEM)/high glucose, containing 10% (*v*/*v*) FBS, 100 U/mL penicillin and 100 µg/mL streptomycin. The 2 × 10^5^ cells were seeded on 0.3 × 0.4 × 0.2 сm^3^ porous scaffold. The cell-seeded scaffolds were then transferred into fresh full culture medium and incubated at +37 °C, 5% CO_2_ for three days. Prior to *K. pneumoniae* UHI 1090 inoculation, the scaffolds with cells were washed three times in DMEM/high glucose medium and incubated in DMEM/high glucose containing 10% FBS overnight at 37 °C, 5% CO_2_. 200 µL of an overnight culture of *K. pneumoniae* UHI 1090 was inoculated into a model system where the final concentration of the bacterial cells was 10^4^ CFU/mL. The incubation lasted at 37 °C under high CO_2_ conditions for two days. Upon incubation, colony forming units (CFU) in the liquid phase of a model were identified by a classical plating assay in triplicate. 

### 4.8. Confocal Laser Scanning Microscopy (CLSM)

Samples taken from the wound model cultures were placed onto glass slides and stained with 2 µg/mL ethidium bromide (ThermoFisher Scientific, Waltham, MA, USA) and 1 µM AmyGreen (Department of Biomedicinal Chemistry, Institute of Molecular Biology and Genetics, Kyiv, Ukraine) in DMSO (Sigma-Aldrich, Saint Louis, MO, USA) without being washed, to limit the physical disruption of biofilm structures through liquid movement. The samples were not fixed, and a cover slip was placed over the stained samples before imaging. CLSM analysis was undertaken using a Leica TCS SPE Confocal system with a coded DMi8 inverted microscope (Leica, Mannheim, Germany) and Leica Application Suite X (LAS X) Version 3.4.1 (Leica Microsystems CMS, GmbH, Wetzlar, Germany). Images were acquired using an excitation wavelength of 488 nm and with emissions collected at 490–580 nm for AmyGreen and excitation at 532 nm and at 537–670 nm for ethidium bromide.

### 4.9. Scanning Electron Microscopy (SEM)

Freeze-dried samples of developed matrices were coated with a 10–30 nm thick gold-metal layer to improve the surface conductivity and examined for morphological details with the Jeol JSM 35C and Jeol JSM 6060LA scanning electron microscopes (Tokyo, Japan).

### 4.10. Statistical Analysis

Replicate data were processed using the statistical software package OriginPro 7.0 (OriginLab Corporation, Northampton, MA, USA), MS Excel for Windows and MaxStat Pro 3.6 (MaxStat Software, Jever, Germany). All results are presented as the mean ± standard deviation. A value of *p* < 0.05 was considered statistically significant.

## 5. Conclusions

An effective combined antibacterial therapy against XDR and PDR *K. pneumoniae* infections which is effective for most of the hospital isolates, can be a good alternative to the synergy tests or other time- and cost-consuming assays which have not been introduced widely into clinic practice. AZM demonstrated effective antibacterial and antibiofilm activities against *K. pneumoniae* UHI 1090 biofilm infections in in vitro wound models based on a 3D collagen scaffold seeded with epithelial-like cell line HEK293. CMS did not have a promising antibacterial effect in combination with AZM, except for a slight improvement as observed in one model. On the other hand, CMS applied alone, either had no effect on the *Klebsiella* population or even stimulated bacterial growth. 

## Figures and Tables

**Figure 1 antibiotics-12-00293-f001:**
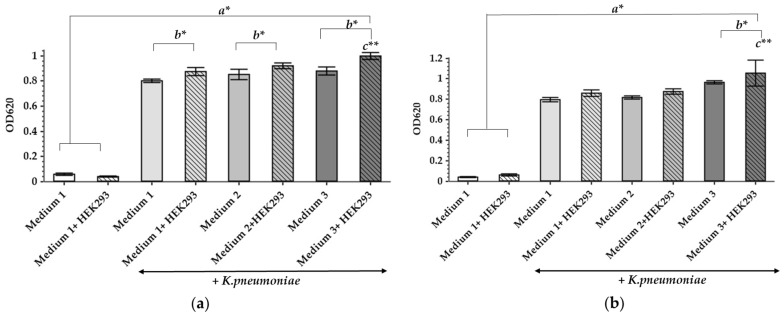
The influence of culture medium and HEK293 cells on the growth of *K. pneumoniae* UHI 1090. Growth shown under static conditions, after 24 h (**a**) and 48 h (**b**) of incubation. Data are expressed as mean values (n = 6). Error bars represent standard deviations. Data were analyzed by one-way ANOVA with a Barlette post hoc test. * Represents significant differences at *p* < 0.05, ** represents significant differences at *p* < 0.001. a–as compared to the control groups incubated without bacteria; b–as compared between the samples incubated with or without HEK293; c–as compared to all other groups.

**Figure 2 antibiotics-12-00293-f002:**
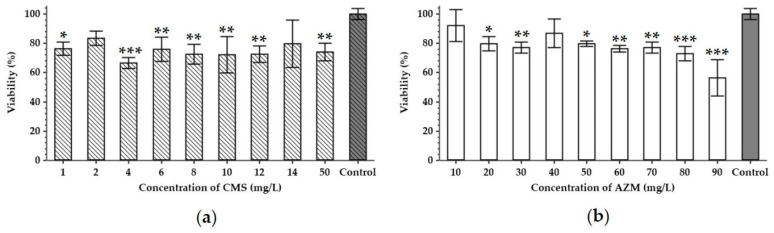
Cytotoxicity of colistin methanesulfonate (**a**) and azithromycin (**b**). The viability of HEK293 (human embryonal kidney) cell line treated with different concentrations of antibiotics for 48 h was analyzed using an MTT assay. Data are expressed as mean values (n = 3). Error bars represent SD. Control represents the HEK293 cultured without the addition of antibiotics. Data were analyzed by the Mann–Whitney U-test; * represents significant differences at *p* < 0.05, ** represents significant differences at *p* < 0.01, *** represents significant differences at *p* < 0.001, each time as compared to the control samples incubated without the addition of antibiotics.

**Figure 3 antibiotics-12-00293-f003:**
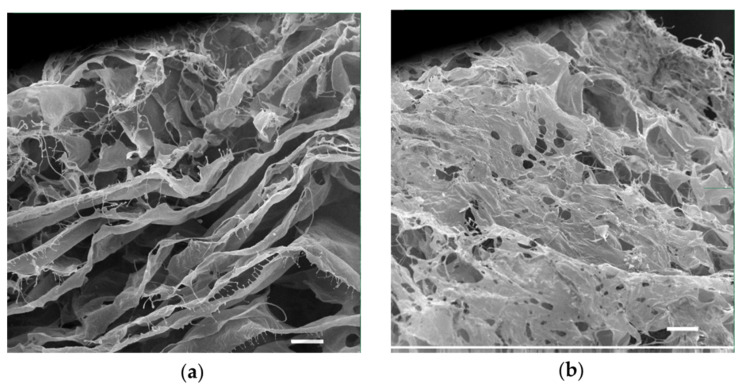
Scanning electron micrographs of a 3D collagen scaffold (scale bar corresponds to 100 µm). (**a**) cross-section; (**b**) the surface of the scaffold.

**Figure 4 antibiotics-12-00293-f004:**
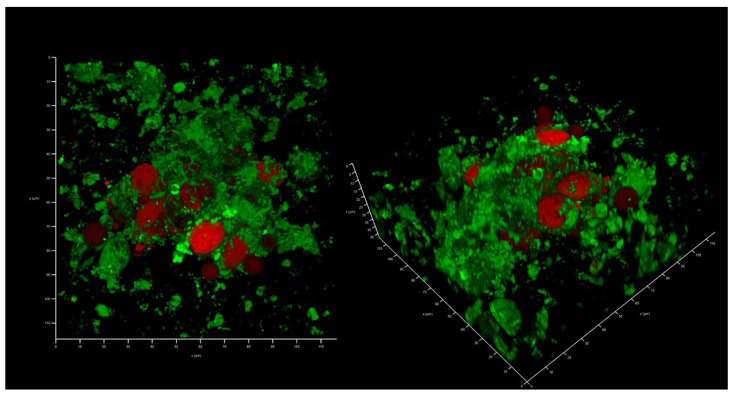
In vitro wound model of a 3D collagen scaffold seeded with HEK293 cells line after three days of incubation. Confocal laser scanning microscopy imaging, 120 µm × 120 µm × 35 µm, 10 µm per division. AmyGreen was used to stain collagen (green signal) and ethidium bromide was used to stain eukaryotic nuclei (red signal).

**Figure 5 antibiotics-12-00293-f005:**
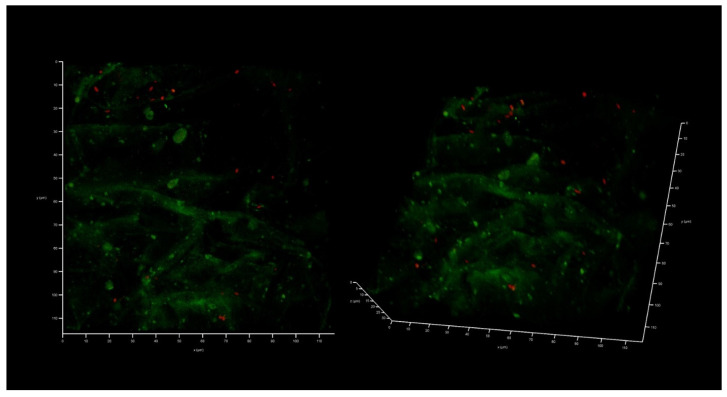
A 48-h old *K. pneumoniae* UHI 1090 culture developed on a 3D collagen scaffold. The scaffold was not seeded with HEK293 cells, but fetal bovine serum was added as a source of blood proteins (medium No. 1). Confocal laser scanning microscopy imaging, 120 µm × 120 µm × 35 µm, 10 µm per division. AmyGreen was used to stain collagen (green signal), and ethidium bromide was used to stain the bacterial cells (red signal).

**Figure 6 antibiotics-12-00293-f006:**
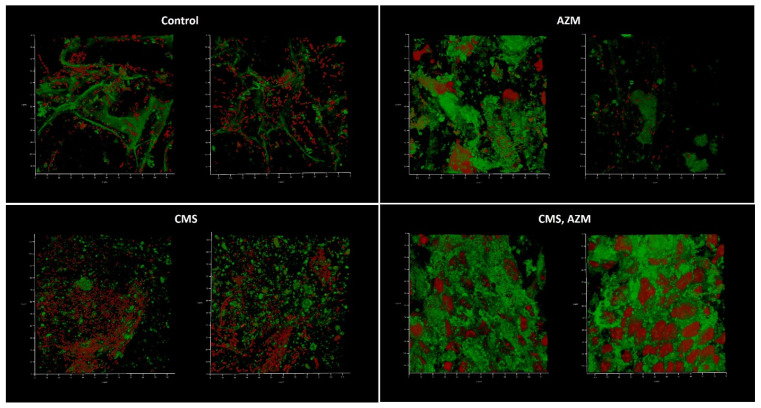
In vitro wound model of a *K. pneumoniae* UHI 1090 biofilm exposed to colistin methanesulfonate and azithromycin. Images were taken after 48 h of biofilm development on a 3D collagen scaffold seeded with the epithelial-like cell line HEK293 in medium No. 1 (90% DMEM/high glucose, 10% fetal bovine serum), supplemented with 5 mg/L colistin methanesulfonate (**CMS**) or 10 mg/L of azithromycin (**AZM**), or both colistin methanesulfonate and azithromycin (**CMS, AZM**), or without (**control**). Confocal laser scanning microscopy imaging, 120 µm × 120 µm × 35 µm, 10 µm per division. AmyGreen was used to stain collagen (green signal) and ethidium bromide was used to stain eukaryotic nuclei and bacteria (red signal).

**Figure 7 antibiotics-12-00293-f007:**
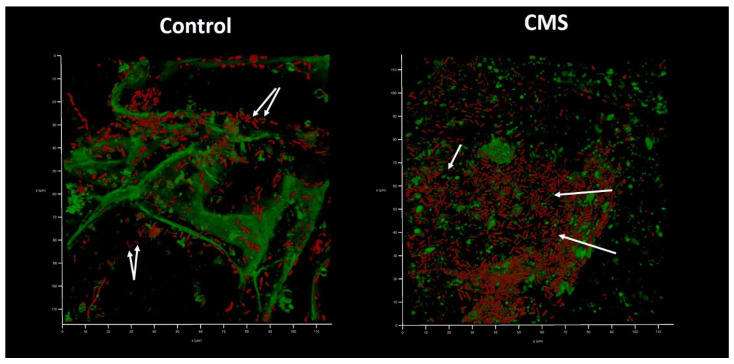
In vitro wound model of a *K. pneumoniae* UHI 1090 biofilm exposed to colistin methanesulfonate (enlarged details of parts of Figure 6). Images were taken after 48 h of biofilm development on a 3D collagen scaffold seeded with the epithelial-like cell line HEK293 in medium No. 1 (90% DMEM/high glucose, 10% FBS) supplemented with 5 mg/L colistin methanesulfonate (**CMS**) or without (**control**). Confocal laser scanning microscopy imaging, 120 µm × 120 µm × 35 µm, 10 µm per division. AmyGreen was used to stain collagen and bacterial amyloid fibers (green signal) and ethidium bromide was used to stain eukaryotic nuclei and bacteria (red signal). Arrows point to amyloid fibers associated with bacterial cells.

**Figure 8 antibiotics-12-00293-f008:**
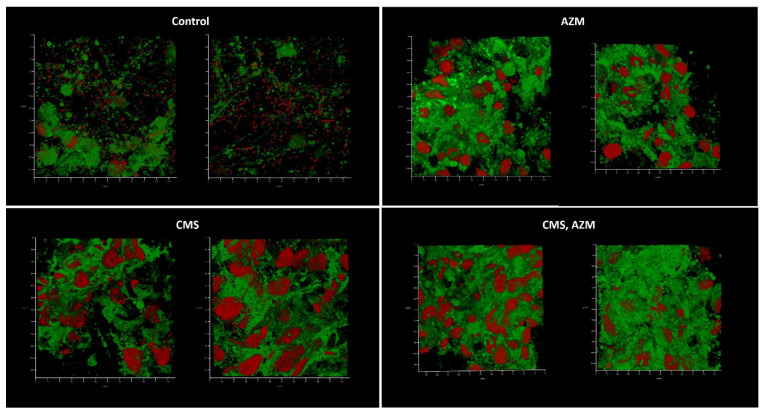
In vitro wound model of a *K. pneumoniae* UHI 1090 biofilm exposed to colistin methanesulfonate and azithromycin. Images were taken after 48 h of biofilm development on a 3D collagen scaffold seeded with the epithelial-like cell line HEK293, in medium No. 2 (89.9% DMEM/high glucose, 10% FBS, 0.01% Bacto Proteose Peptone), supplemented with 5 mg/L colistin methanesulfonate (**CMS**) or 10 mg/L of azithromycin (**AZM**), or both colistin methanesulfonate and azithromycin (**CMS, AZM**), or without (**control**). Confocal laser scanning microscopy imaging, 120 µm × 120 µm × 35 µm, 10 µm per division. AmyGreen was used to stain collagen (green signal) and ethidium bromide was used to stain eukaryotic nuclei and bacteria (red signal).

**Figure 9 antibiotics-12-00293-f009:**
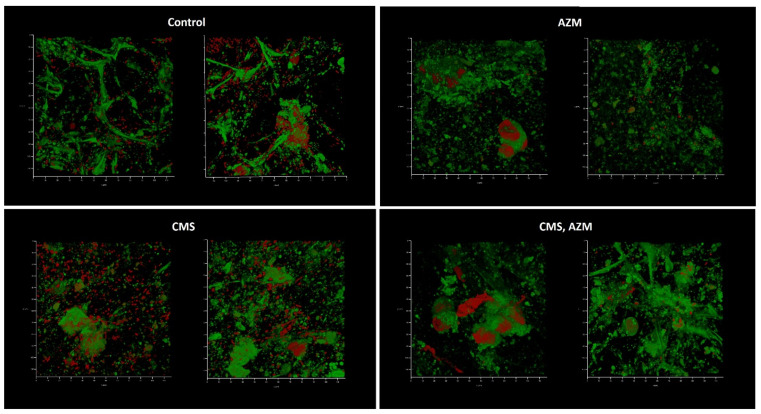
In vitro wound model of a *K. pneumoniae* UHI 1090 biofilm exposed to colistin methanesulfonate and azithromycin. Images were taken after 48 h of biofilm development on a 3D collagen scaffold seeded with the epithelial-like cell line HEK293, in medium No. 3 (49.9% DMEM/high glucose, 50% fetal bovine serum, 0.1% Bacto Proteose Peptone) supplemented with 5 mg/L colistin methanesulfonate (**CMS**) or 10 mg/L of azithromycin (**AZM**), or both colistin methanesulfonate and azithromycin (**CMS, AZM**), or without (**control**). Confocal laser scanning microscopy imaging, 120 µm × 120 µm × 35 µm, 10 µm per division. AmyGreen was used to stain collagen (green signal) and ethidium bromide was used to stain eukaryotic nuclei and bacteria (red signal).

**Figure 10 antibiotics-12-00293-f010:**
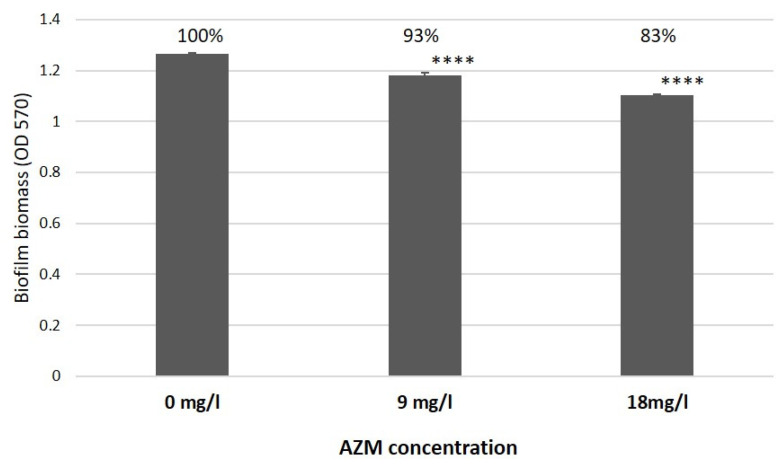
The effect of azithromycin on biofilm growth by *K. pneumoniae* UHI 1090 investigated using biofilm assays and optical density (OD) measurements at 570 nm after 24 h incubation (adapted from Figure 5 in [10]). Statistical differences were determined between media with azithromycin versus control media without; **** *p* < 0.001.

**Figure 11 antibiotics-12-00293-f011:**
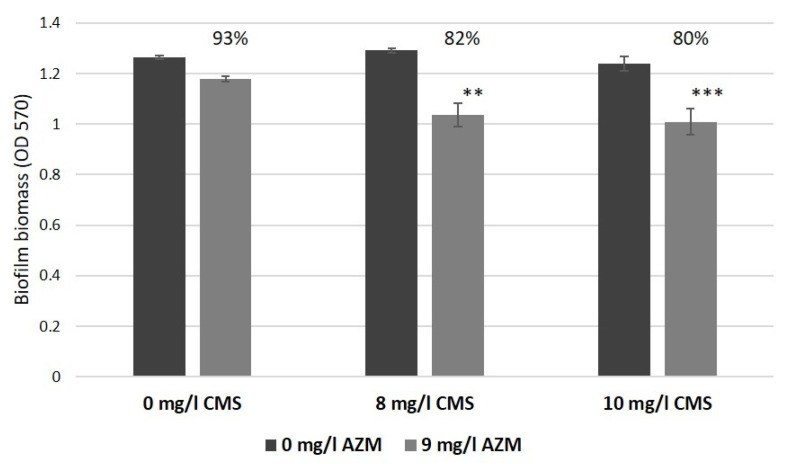
The effect of azithromycin and colistin methanesulfonate on biofilm growth by *K. pneumoniae* UHI 1090 investigated using biofilm assays and with optical density (OD 570) measurements after 24 h incubation (adapted from Figure 5 in [10]). Statistical differences were determined between media with azithromycin versus control media without; ** *p* < 0.01, *** *p* < 0.005.

## Data Availability

All data have been incorporated into this manuscript.

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
