# Peer review of "Therapeutic Potential of an Azithromycin-Colistin Combination against XDR K. pneumoniae in a 3D Collagen-Based In Vitro Wound Model of a Biofilm Infection"

_antibiotics, 2023, doi:10.3390/antibiotics12020293_

Round 1

Reviewer 1 Report

This is a very well-written and comprehensive research article focused on the development of an in vitro wound model of biofilm infection to test drug combinations against K. pneumoniae. I do not have any major issues with the article, the data are clearly presented, the bibliography is extensive, and I do not see any major shortcomings with regard to the methodology applied here. I have only a minor question with regards to Fig 7, which seems a zoom of pictures included in Fig. 6, could authors not provide the same information in Fig. 6 to avoid a repetition of the images included in these two Figures?

Author Response

I have only a minor question with regards to Fig 7, which seems a zoom of pictures included in Fig. 6, could authors not provide the same information in Fig. 6 to avoid a repetition of the images included in these two Figures?

We have tried to add the arrows to fig 6. but deemed the details too small to be seen convincingly. We would in this case defend the repetition of said figures for the sake of clarity.

Reviewer 2 Report

The manuscript presented for review presents an interesting potential therapeutic application of XDR treatment of resistant Gram-negative bacteria.

It is edited correctly and requires only editorial corrections, such as in vitro written in italic (line 3, 29,31,33).

I also suggest moving figures 10 and 11 and their descriptions to the results section.

Author Response

It is edited correctly and requires only editorial corrections, such as in vitro written in italic (line 3, 29,31,33).

Checked in the whole manuscript and adapted – in normal text Latin expressions are now in italics, while in titles (which are completely in italics), they are put straight.

I also suggest moving figures 10 and 11 and their descriptions to the results section.

We have considered this suggestion and decided against it. The data in these figures are not really results of this paper but were published elsewhere and introduced here as a basis for the discussion. We do not want to create the impression that we were publishing data from a previous paper twice, and therefore prefer to leave the graphs in the discussion section and, evidently, cite the original publication.

Reviewer 3 Report

The manuscript from Moshynets et al. developed a the 3D, collagen-based, in vitro wound model of a biofilm infection which allowed to evaluate the antimicrobial therapeutic potential of an azithromycin - colistin combination against XDR K. pneumoniae.

The authors made a good job and presented s very well structured and interesting study. Thus the manuscript deserves be accepted for publication after minor adjustments.

in the abstract, the authors should include a concluding sentence at the end.

Please observe the use of italic for worlds in Latin, such as in vitro…

Please note that is important to use the same writing symbols and units throughout the text. For instance, ‘L’ and “l” are found in the text. Make this uniform.

The authors should provide a more comprehensive title for figures, in particular in the figure 4.

Please provide the full name instead of abbreviation in the figure captions.

Author Response

in the abstract, the authors should include a concluding sentence at the end.

We have added “The wound model proposed here proves therefore to be an effective aid in the study of drug combinations under realistic conditions.”

Please observe the use of italic for worlds in Latin, such as in vitro…

Checked in the whole manuscript and adapted – in normal text Latin expressions are now in italics, while in titles (which are completely in italics), they are put straight.

Please note that is important to use the same writing symbols and units throughout the text. For instance, ‘L’ and “l” are found in the text. Make this uniform.

To the best of our ability, we have changed every l into L. I hope we did not miss any. We also tried to replace every decimal comma with a decimal point.

The authors should provide a more comprehensive title for figures, in particular in the figure 4. Please provide the full name instead of abbreviation in the figure captions.

Most captions have been rewritten and essential full names have been added.